# Causal evidence for the adaptive benefits of social foraging in the wild

Lysanne Snijders [1,2✉], Stefan Krause [3], Alan N. Tump [4], Michael Breuker [3], Chente Ortiz[5], Sofia Rizzi [5], Indar W. Ramnarine[6], Jens Krause[1,5] & Ralf H.J.M. Kurvers [1,4]

Sociality is a fundamental organizing principle across taxa, thought to come with a suite of adaptive benefits. However, making causal inferences about these adaptive benefits requires experimental manipulation of the social environment, which is rarely feasible in the field. Here we manipulated the number of conspecifics in Trinidadian guppies (*Poecilia reticulata*) in the wild, and quantified how this affected a key benefit of sociality, social foraging, by investigating several components of foraging success. As adaptive benefits of social foraging may differ between sexes, we studied males and females separately, expecting females, the more social and risk-averse sex in guppies, to benefit more from conspecifics. Conducting over 1600 foraging trials, we found that in both sexes, increasing the number of conspecifics led to faster detection of novel food patches and a higher probability of feeding following detection of the patch, resulting in greater individual resource consumption. The extent of the latter relationship differed between the sexes, with males unexpectedly exhibiting a stronger social benefit. Our study provides rare causal evidence for the adaptive benefits of social foraging in the wild, and highlights that sex differences in sociality do not necessarily imply an unequal ability to profit from the presence of others.

[1] Department of Biology and Ecology of Fishes, Leibniz-Institute of Freshwater Ecology and Inland Fisheries, 12587 Berlin, Germany. [2] Behavioural Ecology Group, Wageningen University, 6708 PB Wageningen, The Netherlands. [3] Department of Electrical Engineering and Computer Science, Lübeck University of Applied Sciences, 23562 Lübeck, Germany. [4] Center for Adaptive Rationality, Max Planck Institute for Human Development, 14195 Berlin, Germany. [5] Faculty of Life Sciences, Humboldt-Universität zu Berlin, 10115 Berlin, Germany. [6] Department of Life Sciences, University of the West Indies, St Augustine, Trinidad and Tobago. ✉email: snijders@igb-berlin.de

Sociality evolved convergently in a wide range of taxa wherever its benefits (e.g., reduced predation risk and increased foraging success) outweighed its costs (e.g., increased conspicuousness and competition)[1,2]. This cost–benefit ratio of sociality is more variable in some animal populations than others, leading to a variety of social systems, from facultative social populations with a high degree of fission–fusion to obligate social populations with little fission–fusion[3]. Their socially plastic nature makes facultative social populations highly suitable study systems for investigating the costs and benefits of sociality—for example, by correlating natural variation in group size to variation in group performance[4], while including the performance of solitary individuals. However, studies examining natural variation in group size, although highly informative[5–8], do not allow causal conclusions to be drawn on the costs and benefits of sociality for individuals. For example, in observational studies individuals have the opportunity to self-select their social environment, which may lead to hidden correlations in group composition and group size. In addition, both group size and group performance may vary in response to variation in the environment (e.g., resource abundance), making it challenging to elucidate the driving factor. To allow for causal conclusions, experimental studies are conducted that directly manipulate the social environment[9–14]. Yet, these studies are predominantly done in captive settings and, therefore, likely to miss out on ecological relevance as they are unable to incorporate all of the local environmental pressures that shaped the selection for sociality.

Experimental studies that manipulate the number of conspecifics in the wild are few and far between (with the notable exception of removal experiments). Such studies are vital for drawing ecologically and evolutionary relevant conclusions on the costs and benefits of social life. One key benefit of sociality is the opportunity for social foraging[15]. Although the exact benefits of the presence of conspecifics vary with ecological conditions[16,17] (e.g., food abundance[18]), the presence of conspecifics is generally thought to increase the mean (or reduce the variance in) individual foraging performance[15,19]. Due to the challenges of setting up, and subsequently replicating, different social compositions in the field, few studies have been able to experimentally manipulate the social environment in vertebrate species and study its effects on foraging performance in the wild[20–22]. Fewer, if any, have been able to manipulate conspecific number, including representative solitary conditions.

Whether and how much an individual gains from the presence of conspecifics during foraging depends on individual-level and group-level traits that modulate the effectiveness of the underlying social foraging mechanisms. Mechanisms that may underlie positive effects of sociality on individual foraging performance include local and stimulus enhancement[23], decreased neophobia[13], social facilitation[24], social and public information use[25], perceived safety (leading to e.g., a reduction in anti-predation behavior in favor of foraging[11]), perceived competition (leading to e.g., an increase in foraging effort to increase one's resource share[12]), pool of competence[7] and cooperation[5]. Mechanisms that may underlie negative effects of sociality include social attraction away from a profitable resource[26], misleading social information[27], increased interference[28] and exclusion from the resource[29]. Identifying the exact mechanisms underlying social effects is challenging, especially in the wild. However, by studying how the number of conspecifics changes different components of foraging performance (e.g., resource detection and resource acquisition), and whether particular individual traits modulate these changes, we can gain more insight not only into whether social foraging has benefits, but also into the mechanisms underlying these benefits.

Sex is likely to be one of the prime individual-level traits influencing the potential benefit(s) of social foraging. Many individual-level characteristics that are predicted to modulate the effectiveness of social foraging mechanisms, such as social position[30], risk-sensitivity[31], energy requirement[32], and dominance[33], covary with sex[18,31,34–39]. We may thus expect conspecific presence to have a stronger impact on foraging performance in one sex than the other[40,41].

Here, we conducted an in situ manipulation of the number of same-sex conspecifics in a facultative social vertebrate species, wild-living Trinidadian guppies (*Poecilia reticulata*). We varied the number of fish, with either one, four or eight males or females per pool. Subsequently, we conducted food-provisioning trials and quantified different components of individual foraging performance, including (latency to) resource detection, likelihood of feeding following detection (i.e., resource acquisition) and total number of bites (i.e., resource consumption) in seven different natural pools. In our earlier work with wild Trinidadian guppies, living upstream in resource-poor environments, we revealed that more social individuals located more novel food patches[22,42]. We therefore predicted that increasing the number of conspecifics would positively influence individual resource detection. In contrast, due to the lack of predators[43] and the infrequent use of aggression[22] in our population, we did not expect a strong increase (facilitation) or decrease (competition) in resource acquisition with conspecific number. In addition, previous work showed that male guppies are generally less social[22,44–48] and more risk-taking than females[44,49–51]. We therefore predicted a shallower increase in foraging performance, in terms of both resource detection and acquisition, with increasing number of conspecifics for males than for females. Moreover, given females' higher nutritional demands, we expected females to show a stronger foraging performance overall. Finally, to evaluate whether the predicted shallower increase in males' foraging performance could be explained by males generally taking less advantage of the presence of others, rather than by male conspecifics providing less effective social cues to other males, we also tested control compositions of one focal male with up to seven female conspecifics.

With this experimental field study, we provide rare causal evidence for the adaptive benefits of social foraging in the wild. Moreover, we demonstrate that both male and female guppies can gain a foraging advantage from sharing their habitat with an increasing number of conspecifics. Our results thus indicate that established sex differences in sociality do not necessarily imply an unequal ability to benefit from social presence.

## Results
**Resource detection: presence at novel food patches**. Although treatments initially consisted of either one, four or eight fish per pool (Supplementary Table 1), escapes prior to the food-provisioning trials resulted in a more diverse range in the number of fish per batch, especially for females (Supplementary Table 2). We conducted and filmed a total of 1645 food-provisioning trials. For 1619 trials we could reliably quantify whether or not each batch member had been present and for 1559 we could reliably assess the arrival latency and number of foraging bites of each individual. All analyses were conducted using a model selection approach (see "Methods" section). Generalized linear mixed models were employed for all dependent variables with the exception of the arrival latency for which we employed a Cox proportional hazards model with mixed effects (i.e., survival analysis).

Using generalized linear mixed model selection, we found that a larger number of fish in a pool, independent of sex, increased the chance and speed of a novel food patch being discovered by any fish (Supplementary Note 1), suggesting the presence of

**Table 1 Generalized linear mixed model summary for presence at a novel food patch.**

| Predictor variable | Estimate | SE | $\chi^2$ | p |
|---|---|---|---|---|
| Number of conspecifics | **0.22** | **0.07** | **10.27** | **0.001** |
| Sex | −0.11 | 0.14 | 0.61 | 0.43 |
| Body length | 0.06 | 0.05 | 1.73 | 0.19 |
| Trial number | 0.06 | 0.03 | 2.77 | 0.096 |
| Pool surface area | −0.02 | 0.13 | 0.02 | 0.88 |
| Sex: Number of conspecifics | 0.02 | 0.13 | 0.01 | 0.91 |
| Sex: Body length | −0.06 | 0.09 | 0.36 | 0.55 |

Presence was analyzed using a binomial error distribution. Individual identity nested in batch identity, batch identity and patch location identity nested in pool identity were added as random effects. Continuous variables were scaled and body length was centered to sex. We used stepwise backward model selection, assessing the significance of the change in deviance upon removal of the effect, using log-likelihood ratio tests. Estimates are reported for the last model still including the effect. Significant effects are reported in bold. Number of observations = 5070.

**Table 2 Cox proportional hazards mixed model summary for arrival latency at a novel food patch.**

| Predictor variable | Hazard ratio | CI | $\chi^2$ | p |
|---|---|---|---|---|
| Number of conspecifics | **1.17** | **1.06–1.30** | **9.78** | **0.002** |
| Sex | 0.88 | 0.70–1.09 | 1.43 | 0.23 |
| Body length | 1.04 | 0.97–1.12 | 1.25 | 0.26 |
| Trial number | 1.04 | 0.99–1.10 | 2.43 | 0.12 |
| Pool surface area | 0.94 | 0.76–1.16 | 0.36 | 0.55 |
| Sex: Number of conspecifics | 1.06 | 0.87–1.30 | 0.36 | 0.55 |
| Sex: Body length | 0.91 | 0.79–1.05 | 1.60 | 0.21 |

Trials in which individuals did not arrive at the food patch were assigned a latency of 120 s and labeled as right-censored. Individual identity nested in batch identity, batch identity, patch location identity nested in pool identity and pool identity were added as random effects. Continuous variables were scaled and body length was centered to sex. Hazard ratios were calculated by 'exp(coef)'. We used stepwise backward model selection, assessing the significance of fixed effects by the change in deviance upon removal of the effect, using log-likelihood ratio tests. Estimates are reported for the last model still including the effect. Significant effects are reported in bold. Number of uncensored observations = 1423 and total observations = 4841.

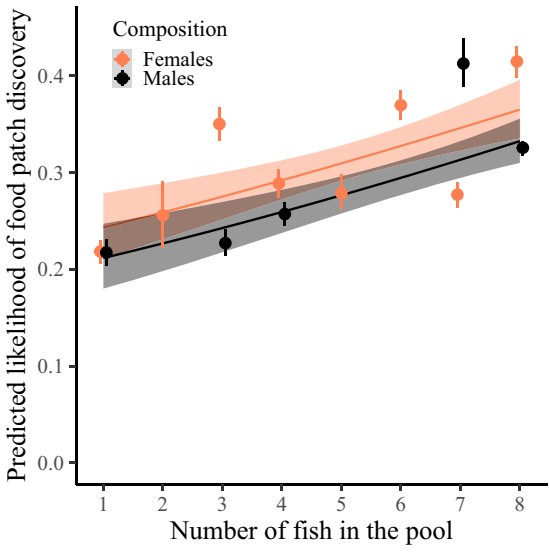

**Fig. 1 Predicted likelihood of novel food patch discovery as a function of the number of fish in the pool.** Dots with bars represent the mean ± 95% confidence interval (CI) predicted value summary statistics for each sex (female/male = orange/black) and fish number (CI obtained from 1000 bootstraps). Grouping of the data was conducted for graphical purposes only; analyses were conducted on an individual-by-trial level (n = 5070) with the number of fish in the pool as a continuous variable (Table 1). Note that there were no batches with two, five or six males (Supplementary Table 2). Regression lines show the predicted final model values. Shaded areas around the lines reflect 95% CI. A slight horizontal position dodge was added to reduce overlap.

advantageous social information that individuals could exploit to improve their foraging performance. Indeed, the higher the number of conspecifics in the pool, the more likely individuals were to discover a novel patch (Table 1; Fig. 1). Males and females benefited similarly from the presence of more same-sex conspecifics (Table 1; Fig. 1) and did not differ in their overall likelihood of finding a novel food patch (Table 1; Fig. 1). Body length did not affect the likelihood of novel food patch discovery (Table 1) in either males or females (Table 1). There was a non-significant tendency for the likelihood of novel food patch discovery to increase with trial number (Table 1). There was no effect of pool surface area (Table 1).

**Resource detection: arrival latency at novel food patches.** Individuals with more conspecifics in the pool were also quicker to reach a novel food patch and this positive social effect was, again, similar for males and females (Table 2; Fig. 2). Males and females also did not differ in how quickly they reached a novel food patch (Table 2; Fig. 2). Body length did not affect the speed of reaching a novel food patch in either males or females (Table 2). Trial number did not affect how quickly an individual reached a novel food patch and neither did pool surface area (Table 2). Also when looking at recruitment latency (defined as the time between the arrival of the first and second fish), we found no difference between males and females (Supplementary Note 1).

**Resource acquisition: motivation to feed from novel food patches.** Individuals with more conspecifics in the pool were more likely to take a bite while present at a food patch, and this positive effect of conspecifics on the motivation to feed was stronger for males than for females (Table 3; Fig. 3). Males were less likely than females to take a bite when they were solitary (Estimate (Est) ± SE = −4.24 ± 1.57, n = 126, $\chi^2$ = 13.36, p < 0.001) but tended to be more likely than females to feed when they were with more than five fish in the pool (Est ± SE = 1.68 ± 0.92, n = 966, $\chi^2$ = 3.19, p = 0.074; Fig. 3). Body length influenced whether an individual would feed, with a negative effect in males and a positive effect in females (Table 3; Supplementary Fig. 1). There was no effect of trial number on the likelihood of an individual taking a bite at a food patch, nor was there an effect of pool surface area (Table 3).

**Resource consumption: total number of bites.** Individuals with more conspecifics in the pool took more bites, with males showing a stronger increase with the number of conspecifics than females (Table 4; Fig. 4). Solitary males took fewer bites than solitary females (Est ± SE = −3.05 ± 1.15, n = 31, $\chi^2$ = 14.39, p < 0.001; Fig. 4), but males and females in a pool with at least six other male or four other female conspecifics did not differ (Est ± SE = 0.49 ± 0.47, n = 167, $\chi^2$ = 1.11, p = 0.29; Fig. 4). The number of bites decreased with body length in males, but increased with body length in females (Table 4; Supplementary Fig. 2). Pool surface area had no effect on the total number of bites per individual (Table 4).

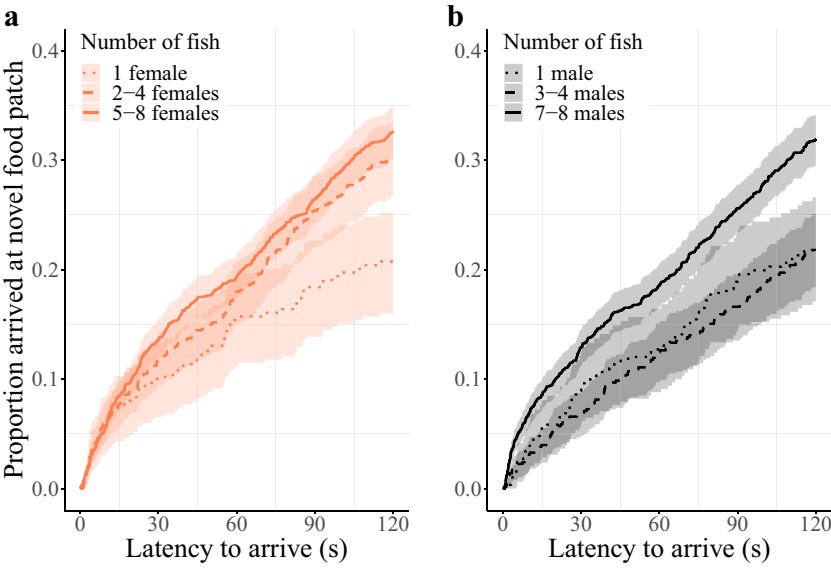

**Fig. 2 Proportion of observations in which individuals arrived at the novel food patch as a function of time.** Higher values on the y-axis reflect an increasing number of individuals arriving at a food patch by that time. Grouping of the number of fish in the pool for **a** females (orange) and **b** males (black) was conducted for graphical purposes only (solitary = dotted line, 2–4 fish = dashed line, 5–8 fish = solid line); analyses were conducted on an individual-by-trial level (*n uncensored/n total* = 1423/4841) with fish number as a continuous covariate (Table 2). Note that there were no batches with two, five or six males (Supplementary Table 2). Regression lines show the predicted values of a simplified model (excluding additional co-variates and random effects). Shaded areas around the lines reflect 95% confidence intervals.

**Table 3 Generalized linear mixed model summary for feeding at a novel food patch while present.**

| Predictor variable | Estimate | SE | $\chi^2$ | p |
|---|---|---|---|---|
| Trial number | 0.06 | 0.09 | 0.45 | 0.50 |
| Pool surface area | −0.09 | 0.46 | 0.04 | 0.84 |
| Sex: Number of conspecifics | **1.95** | **0.48** | **16.73** | **0.0004** |
| Sex: Body length | **−1.23** | **0.39** | **10.49** | **0.001** |

Feeding (taking at least one bite) was analyzed using a binomial error distribution. Individual identity nested in batch identity, batch identity, patch location identity nested in pool identity and pool identity were added as random effects. Continuous variables were scaled and body length was centered to sex. We used stepwise backward model selection, assessing the significance of fixed effects by the change in deviance upon removal of the effect, using log-likelihood ratio tests. Estimates are reported for the last model still including the effect. Significant effects are reported in bold. Statistics for main effects of variables in significant interactions are omitted. Number of observations = 1423.

**Foraging performance of males in the presence of females.** Consistent with our previous studies[22,42], there was a trend for males with male conspecifics in the pool to be less likely to reach a novel food patch than males with female conspecifics, i.e., in control compositions (Est ± SE = −0.60 ± 0.35, $n$ = 1828, $\chi^2$ = 2.79, $p$ = 0.09; Supplementary Fig. 3). However, sex of the conspecifics in the pool did not affect how quickly males reached a novel patch (Hazard ratio (CI) = 0.67 (0.40–1.13), $n$ not censored = 572, $n$ total = 1713, $\chi^2$ = 2.20, $p$ = 0.14; Supplementary Fig. 4). Males with male conspecifics in the pool were, in contrast, more likely to bite when present at a food patch than males with female conspecifics (Est ± SE = 2.32 ± 0.97, $n$ = 572, $\chi^2$ = 5.02, $p$ = 0.02; Supplementary Fig. 5), resulting in a non-significant tendency for males with male conspecifics to also gain more bites in total (Est ± SE = 0.78 ± 0.44, $n$ = 97, $\chi^2$ = 3.24, $p$ = 0.07; Supplementary Fig. 6).

## Discussion

The costs and benefits of social living are a central research topic in ecology and evolution, yet few studies have been able to manipulate key modulators of this cost–benefit trade-off in the field. The present manipulation of conspecific presence in a facultatively social fish population provides causal evidence for an increase in resource detection (faster, more frequent), resource acquisition (more likely to feed) and total resource consumption (more bites) with an increasing number of conspecifics for both male and female guppies in the wild.

In many vertebrate species, there are strong sex differences in social tendency and social interest[31,34,35,38,39,52] that may already be present from an early age[53]. In this species, males spend less time near same-sex conspecifics[22], are more likely to leave shoals[46] and are less likely to form stable cooperative bonds[45,48] than their female conspecifics. Our finding that males derived equal, if not greater, benefit as females from social foraging demonstrates that social foraging mechanisms are not necessarily less effective in classes of less social individuals (but see ref. [40]), possibly because some of these social mechanisms are not an adaptation to social life per se[54]. Indeed, even non-social species, such as the solitary-living red-footed tortoise (*Geochelone carbonaria*), are capable of using social cues to their advantage[55]. Comparisons between grouping and non-grouping fish species have also found no differences in social information use[56]. We built upon these previous findings by showing that, also within species, classes of less social individuals can use such social mechanisms equally well and, most importantly, that they use this capability to gain an advantage in a fitness-determining context in the wild.

We predicted that in both solitary and social conditions, females would show stronger foraging performance due to increased selective pressure from the strong link between resource availability and female fecundity[57–59], and given our earlier findings[22]. However, although females outperformed males in the solitary condition, males reached comparable levels of total resource consumption when at least six other same-sex conspecifics were present. Higher interference competition at the patch[28] may have partly constrained fish to take more bites and consequently prevented females to outperform males in highly social conditions. We can only speculate as to which

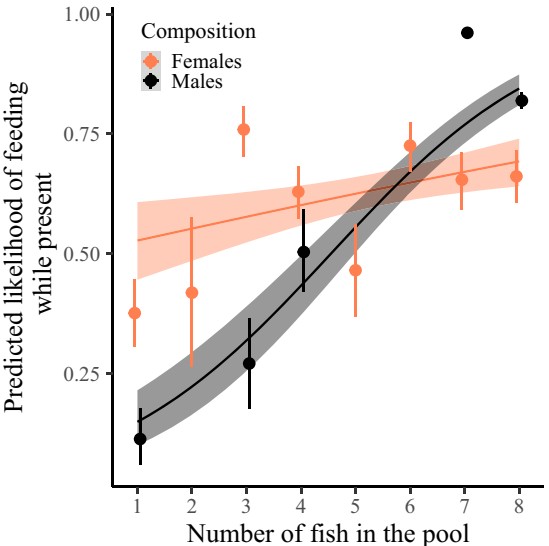

**Fig. 3 Predicted probability of feeding when at a food patch as a function of the number of fish in the pool.** Dots with bars represent the mean ± 95% confidence interval (CI) predicted value summary statistics for each sex (female/male = orange/black) and fish number (CI obtained from 1000 bootstraps). Grouping of the data was conducted for graphical purposes only; analyses were conducted on an individual-by-trial level (n = 1423) with the number of fish in the pool as a continuous variable (Table 3). Note that there were no batches with two, five or six males (Supplementary Table 2). Regression lines show the predicted final model values. Shaded areas around the lines reflect 95% CI. A slight horizontal position dodge was added to reduce overlap.

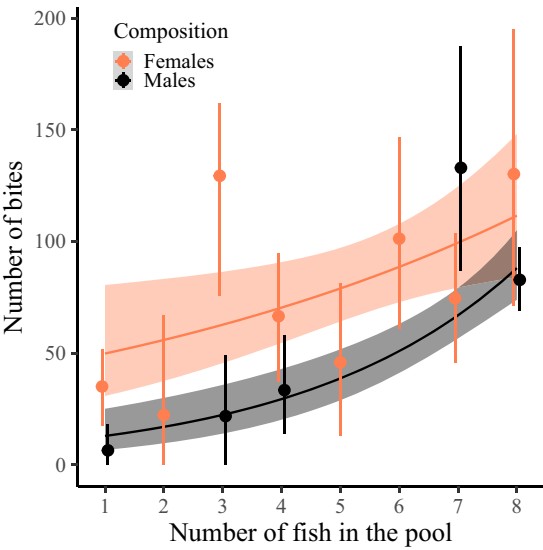

**Fig. 4 Number of bites per individual as a function of the number of fish in the pool.** Dots with bars represent the mean ± 95% confidence interval (CI) summary statistics for each sex (female/male = orange/black) and fish number (CI obtained from 1000 bootstraps). Grouping of the data was conducted for graphical purposes only; analyses were conducted on the level of individuals (n = 265) with the number of fish in the pool as a continuous variable (Table 4). Note that there were no batches with two, five or six males (Supplementary Table 2). Regression lines show the predicted final model values. Shaded areas around the lines reflect 95% CI. A slight horizontal position dodge was added to reduce overlap.

**Table 4 Generalized linear mixed model summary for total bites taken per individual.**

| Predictor variable | Estimate | SE | $\chi^2$ | p |
|---|---|---|---|---|
| Total number of trials | 0.12 | 0.08 | 2.39 | 0.12 |
| Pool surface area | −0.02 | 0.24 | 0.01 | 0.93 |
| Sex: Number of conspecifics | **1.07** | **0.29** | **13.35** | **0.0003** |
| Sex: Body length | **−0.29** | **0.14** | **4.32** | **0.04** |

Number of bites per individual was analyzed using a negative binomial error distribution. Individuals that were never present at any patch (n = 6) were assigned zero bites. Batch identity and pool identity were added as random effects. Continuous variables were scaled and body length was centered to sex. We used stepwise backward model selection, assessing the significance of fixed effects by the change in deviance upon removal of the effect, using log-likelihood ratio tests. Estimates are reported for the last model still including the effect. Significant effects are reported in bold. Statistics for main effects of variables in significant interactions are omitted. Number of observations = 265.

mechanism(s) generated the strong positive social effect in males, but given that the increase in the number of bites appears to be driven mostly by an increase in the probability of feeding when present, mechanisms such as local enhancement[23], pool of competence[7] and cooperation[5] can be excluded. Perceived competition, which is predicted to increase feeding rate[12], is also an unlikely explanation as we would not expect it to change the motivation of fish that are not feeding at all. Increase in perceived safety[11] or decrease in neophobia[13] are also unlikely mechanisms, as we then would expect females (the more risk-averse and possibly more neophobic sex in guppies[44,49–51]) to show the strongest improvement in social versus asocial conditions. Alternatively, solitary males may have underperformed in resource acquisition because they were (also) looking for females, while males in the

social condition may have used the presence of other males as an indicator that females should already be present[60]. Males in the company of several other foraging males may have been more motivated to linger and feed, making social facilitation[24,61] a likely mechanism of the observed increase in total resource consumption. This explanation is also in line with the socio-ecological theory that predicts that female distribution is governed primarily by the distribution of resources and risk, whereas male distribution is governed primarily by the (presumed) distribution of females[59,62,63].

To maximize their fitness, many male animals have to ensure survival via sufficient resource consumption, while at the same time being prolific reproducers. Given that female presence is strongly linked to the presence of resources[59,62,63], males frequently have to choose between courting and foraging. An earlier study suggests that male Trinidadian guppies make state-dependent trade-offs in such situations, ensuring first that they have sufficient energy reserves, but then making a noticeable switch to courtship[64]. Indeed, although males tended to reach more novel food patches when in the presence of females, consistent with our previous findings[22,42], they were less likely to feed than males in single-sex compositions, possibly because they were pursuing mating opportunities. Our results thus highlight the relevance not only of conspecific number, but also of sex-composition, for shaping individual foraging performance in the wild. Future studies could experimentally lower the energy reserves of individual males. e.g., by temporarily restricting access to food, and examine whether the motivation to feed in mixed compositions approaches the level of that of males in single-sex compositions, so as to test whether the influence of composition on individual foraging performance is indeed mainly state-dependent.

The natural local environment that individuals inhabit modulates the cost–benefit ratio of sociality. Conducting experiments

within the natural local environment of the study population, keeping natural selection pressures mostly intact, ensures that the findings will be maximally ecologically relevant. For example, when local food resources are limited, associating in larger shoals may speed up resource detection[65], which is especially beneficial in environments in which solitary detection of food resources is time consuming[66]. This benefit of sociality may be strengthened or outweighed by effects of local predation pressure, with increasing group size leading to less (e.g., "many-eyes" theory, dilution or confusion effect) or more (e.g., increased conspicuousness) individual-level predation risk[67–69]. Taking a comparative approach and conducting this experiment across populations inhabiting different environments[70] would be an intriguing next step to identify the ecological preconditions that allow individuals to benefit from foraging socially—and hence make socially mediated foraging success a relevant contributor to the promotion of sociality.

In conclusion, we experimentally demonstrated the positive effect of conspecific presence on individual foraging performance in the wild. Moreover, by showing that both sexes were able to reach similar foraging performance levels when in the presence of others, we increased our understanding of the individual traits that may shape the evolution of sociality through effects on individual social foraging performance. That male guppies are generally, but not always[22,42], found to be less social, suggests that foraging benefits may not outweigh the costs of being social for males or, alternatively, that we as researchers may need to be more precise in our definition of 'social' (e.g., preferred associations, time spent social or number of unique social contacts). In addition, the documented sex differences in guppy sociality may be mainly driven by females following sex-specific benefits outside the foraging context (e.g., avoiding predation). Investigations into the individual states and ecological characteristics that modulate the individual costs and benefits of sociality in the wild, both within a foraging context and beyond, offer fruitful avenues for future research.

## Methods

**Study system.** We conducted the study between 4 and 20 March 2018 in the upper rainforest region of the Turure River in the Northern Range of Trinidad and Tobago (10°41'8"N, 61°10'22"W). This site has relatively few guppy predators[43,70] and is relatively resource poor due to low sunlight exposure[71]. We used seven natural pools in which we rearranged rocks and pebbles to maintain continuous water flow while minimizing the risk of fish migration. The average surface area of these pools was 3.3 m² (range: 2.4–4.6 m²; Supplementary Table 1); the average depth, based on five measurements in each pool, was 0.16 m (range 0.12–0.26 m; Supplementary Table 1). Guppies originally occurring in the pools were taken out. Experimental fish (only adults) were caught from a nearby stretch of the same river and were, upon capture, sexed (194 females, 143 males), sized (females: Mean ± SD = 24.6 ± 3.7 mm, males: Mean ± SD = 21.6 ± 1.6 mm) and individually marked using Visible Implant Elastomer (VIE) tags (©Northwest Marine Technology Inc.)[22,47,72]. The research complied with all the relevant ethical regulations for animal testing and research at the time and in the country of study (Trinidad and Tobago). We performed all research in accordance with the 'Basic Principles Governing the Use of Live Animals and Endangered Species in Research at the University of the West Indies' as part of the 'Policy and Procedures on Research Ethics' of the University of the West Indies Committee on Research Ethics. All subjects were released back into the stream of capture upon completion of the food provisioning trials.

**Experimental treatments.** We assigned subjects to single-sex batches of one, four or eight fish (main treatments) or to a mixed-sex batch of one male and seven females (control treatment). Due to fish escaping (14% of 392 assigned individuals; with females being more likely to escape (41/235) than males (14/157): $\chi^2 = 4.99$, $p = 0.03$), we ultimately had a more diverse range of group sizes (females: one to eight fish; males: one, three, four, seven and eight fish; control: five to eight fish), spread over 84 batches (39 female batches, 35 male batches, 10 control batches; Supplementary Table 2). The different treatments were approximately balanced over the seven pools according to an a-priori determined schedule so that each pool received each of the seven treatments one to three times (Supplementary Table 1). Following marking, fish were placed in their designated pool and left overnight to acclimate. Foraging trials took place the next day. After finishing the foraging trials, we released subjects further downstream (to avoid recapture).

**Foraging trials.** Foraging trials were conducted following a protocol similar to prior work[22,42]. We assigned five feeding locations, roughly equidistant from one another, in each pool, to offer novel food to the guppies. To standardize the food presentations, each location was marked by an opaque plastic cylinder (diameter: 77 mm, height: 30–40 mm) floating on the surface and kept in place by two wooden skewers. Through these cylinders, we lowered a food item consisting of a small lead ball (diameter: 8 mm) covered in a mix of gelatine and fish food (TetraPro©; Spectrum Brands Inc), which was attached to a monofilament fishing line connected to a wooden rod. The fish food included carotenoids, an important dietary component for guppies[73,74]. We kept the food item (termed 'food patch' from here on, as several fish could feed from it simultaneously) approximately two centimeters above the bottom of the pool for 2 min, irrespective of whether and when it was discovered. After each trial, we waited for 1 min before starting a new trial in a different location. Once we had completed trials for all five locations of a pool in random order, we repeated this procedure three more times, resulting in 20 trials per batch, with some exceptions due to rain, leading to 1645 trials in total. Sample size was informed by our prior work[22,42], but no formal sample size calculation was conducted given that no prior study experimentally manipulated number of conspecifics, including solitary individuals, to evaluate foraging success in wild guppies.

**Video analyses.** We recorded all foraging trials with camcorders (SONY HDR-PJ530E), mounted on tripods. Two observers analyzed the recordings using BORIS v 7.5[75], a free open-source event-logging software. The two observers analyzed different sets of trials, but both sets included all seven treatments and all seven pools. For each fish, the observer scored its presence, arrival latency and number of foraging bites. We defined a fish as present when it was within two body lengths of the food patch. To test the inter-observer reliability, we had both observers score the trials for the same set of six batches (30 unique individuals). The scores for arrival latency, total number of trials present and total number of foraging bites all correlated strongly between the two observers ($r_s > 0.9$). For all trials, the food discovery latencies and presence/absence of individuals were also compared with field notes. In case of discrepancy, the video was checked again and, if necessary, scores were amended by a third observer who had also been present in the field. Not all videos could be reliably analyzed (e.g., due to glare), leading to a small variation between analyses in the number of included trials ($n = 1559–1619$). Video observers were not informed about the study hypotheses.

**Statistics and reproducibility.** To investigate whether the number of conspecifics in the pool increased individual foraging performance (e.g., via social facilitation or social information use) and whether the strength of this effect was sex-dependent, we tested the effects of fish number (number of guppies in the pool), sex-composition (sex of the guppies in the pool: all-male or all-female, excluding control batches) and their interaction on four response variables: whether an individual found a patch (yes/no), for each trial (model I, $n = 5070$); the arrival latency of an individual, for each trial (model II, $n = 4841$); whether an individual took at least one bite from a patch (yes/no), for each patch visited (as a measure of motivation to feed; model III, $n = 1423$) and, finally, the total number of foraging bites per individual across all trials (model IV, $n = 265$). Detailed information on the number of replicates, both on batch-level and individual-level can be found in the 'Experimental treatments' section and Supplementary Tables 1 and 2.

We used stepwise backward model selection, assessing the significance of fixed effects by the change in deviance upon removal of the effect, using log-likelihood ratio tests. Next to the interaction and main fixed effects of fish number (integer, scaled) and sex-composition (factor), all starting models included the main effect of pool surface area (continuous, scaled) and the main effects of body length in mm (integer, scaled and centered on sex) and its interaction with sex-composition. Batch identity and pool identity were included as random effects (Supplementary Fig. 7). All models—except model IV—further included the fixed effect of trial number (integer, scaled) and the random effects of individual identity (nested in batch identity) and patch location identity (nested in pool identity; Supplementary Fig. 8). Model IV uniquely included the number of trials conducted as a fixed effect (integer), which was kept in the model at all times to account for the slightly varying number of trials—and thus varying foraging opportunities—between individuals. Fish number and sex-composition were kept in the models at all times, irrespective of their significance, since they were our fixed effects of interest. Interactions and fixed effects with $P > 0.1$ were removed (unless stated differently above), starting with the least significant interaction followed by the least significant main effect. Estimates are reported for the last model still including the effect. In the case of a significant interaction between fish number and sex-composition, we additionally ran the respective model including only the singleton or the >4 fish treatments in order to specifically investigate potential sex differences in solitary versus social foraging performance.

Model selection was conducted with R version 4.0.2[76] in R Studio version 1.2.5033 (©2009–2019 RStudio, Inc.). All the data and code generated and analyzed during this study are available in the Open Science Framework repository[77]. Models I and III (binary dependent variable) were analyzed with generalized linear mixed models (GLMM) with a binomial error distribution and logit link function using the glmer function from the 'lme4' package[78], fitted by maximum likelihood (Laplace approximation) using the bobyqa optimizer. Model II (continuous dependent variable) was analyzed with mixed effects repeated measures Cox proportional

hazards models using the 'coxme' package[79], fitted by maximum likelihood. Trials in which individuals did not arrive at the food patch were assigned a latency of 120 s and labeled as right-censored (i.e., '120+') using the Surv function in the 'survival' package[80]. We evaluated the proportional hazards assumption by using the cox.zph function and graphically inspecting the survival curves. Hazard ratios were calculated by exp(coef). Finally, model IV (integer dependent variable), due to over-dispersion, was analyzed by running a GLMM using Template Model Builder (TMB) with a negative binomial error distribution and a log link function, using the glmmTMB function from the 'glmmTMB' package[81]. Individuals that were never present at any patch ($n = 6$) were assigned zero bites.

To evaluate whether any observed sex difference in social foraging performance may have been driven by males having male foraging companions[22], rather than by males generally being poorer (or more proficient) social foragers, we compared the foraging performance of males in the company of other males to that of ten 'control' males in the company of only females. We again used the statistical procedures described above, but excluding the interaction effects, the female-only treatments and treatments with originally fewer than eight fish assigned. All statistical tests are two-sided. Figures were created using the 'ggplot2' package[82].

**Reporting summary**. Further information on research design is available in the Nature Research Reporting Summary linked to this article.

## Data availability
All the data generated and analyzed during this study are available in the Open Science Framework repository, https://osf.io/csajg[77].

## Code availability
The R script with all the code used during this study is available in the Open Science Framework repository, https://osf.io/csajg[77].

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

## Acknowledgements
We are very grateful to Sergio Garcia for helping us with the data collection. The work was funded by an IGB Postdoc Fellowship (2017–2018) and a NWO-Veni Fellowship (2020–current; VI.Veni.192.018) to L.S.

## Author contributions
L.S., R.H.J.M., S.K., and J.K. designed the study. L.S., R.H.J.M., S.K., A.N.T, M.B., and J.K. collected data, C.O. and S.R. extracted data from the videos, L.S. analyzed the data; L.S., I.W.R., R.H.J.M.K., and J.K. contributed materials and infrastructure to the study; L.S. wrote the first draft of the manuscript; and all authors provided feedback to revisions.

## Funding

## Competing interests
The authors declare no competing interests.
