## [Peer Review File · Communications Biology]

REVIEWERS' COMMENTS:

Reviewer #1 (Remarks to the Author):

Review of Snijders et al. "Causal evidence for the adaptive benefits of social foraging in the wild"
I very much enjoyed reading this interesting manuscript and believe it would also be of board interest to biologists in general. Specifically, studies considering the costs and benefits of social foraging probably number in the thousands now, but nearly all of this work is either observations in the wild or captive-based experiments. So, to address this topic with a large-scale experiment like this in the wild will certainly now attract a lot of interest, and be an extremely useful addition to the literature. I found the writing to be excellent, the methods to a great combination of maintaining an intuitive approach to analysing the data as well as being intuitive and understandable for the reader, and the conclusions to follow well from the patterns presented in the data. The supplementary information is clearly formatted and referred to appropriately throughout the main text, and therefore will be useful for readers interested in specific aspects. As such, the suggestions below are aimed mainly at just bringing some attention to a few specific areas which could potentially be considered for some rather straight-forward changes:

L24: Change "...the more social and risk-averse sex,..." to "...the more social and risk-averse sex in this species,..."

L27: I generally try to stay away from referring directly to specific statistical aspects within the abstract, and think it is often more intuitive to use descriptive terms, so I would suggest change "The slope of the latter relationship" to "The extent of the latter relationship"

L40-42: This could be explained a bit further (e.g. an example of why this is the case, or when it has been the case in previous research)

L98: Change "..., indicating..." to "...,potentially indicating..."

L102 Onwards: Probably no need to keep reiterating 'Est \pm SE' (as I think the statement in line 101 will mean that readers will grasp that this is what the first two values separated by ' \pm ' are referring to)

L120-132: "Nnot censored/Ntotal = 1423/4841" is identical all the way through this paragraph and repeated exactly the same 8 times.

L135: Figure 2 is missing the 'A' and 'B' labels and the figure legend is missing info that 'A' is for females and 'B' is for males. Also, it isn't clear by this point in the MS why you'd graphically group females by different number groupings compared to the male number grouping size (i.e. Females 2-4 & 5-8 while Males is 3-4 & 7-8), so could benefit from a brief clarification in the legend.

L164-166: The pattern that the relationship between 'number of conspecifics' and 'bites taken' is stronger for males than females is interesting and clearly very strong in the data. But, I'm just curious whether this pattern is as strong if solitary individuals are excluded from this part of the analysis? Or is it strongly driven by the difference in solitary?

L205-206: Yes, very cool finding! Even within our single population of great tits (*Parus major*), males'

personality is consistently related to various types of social behaviour, including network position (Aplin et al., 2013) and social pair-bonding strength (Firth et al., 2018) during winter foraging, and social assortment in territories during breeding (Johnson et al., 2017), while females' personality is not related to any of these components of sociality. Not sure any of this information will be useful to the authors (indeed, probably more relevant to keep their own examples focussed on fish) but thought I should share as its really interesting to see such obvious effects with sex differences here in their experimental system!

L232-233: This the statement meant generally across species? Or specific to this system/species?

L241-242: It is a bit hard to imagine what exactly is meant by "experimentally lower the energy reserves of individual males" in terms of the proposed methodology. Have any previous studies been successful in this kind of manipulation?

L291: Change "protocol similar to [22,42] " To "protocol similar to prior work [22,42]"

L316-318: These sample sizes are quite important and should probably be mentioned in the earlier 'Results' section in the appropriate sections if there is room to do so.

I hope these comments are useful for the authors, and I congratulate them on an excellent piece of research.

Best wishes,

Josh Firth (please note, I sign all my reviews)

Reviewer #2 (Remarks to the Author):

I found this to be a very interesting and nicely written paper with an impressive dataset to back it up. Manipulating the size of groups in the wild and then recording detailed foraging data is a good way to get at some of the key questions about group foraging.

I think the Results section needs a little bit more methodological detail given the format. I also think tables would be more readable than presenting the results in sentences. Other than this, just a few minor comments/suggestions.

Line 46: This is possibly needlessly picky, but would you include removal experiments here? I know that these probably do not subsequently focus on foraging behaviour, but I feel they somewhat fall into the category of manipulating number of conspecifics in the wild.

Line 95: In this sort of results first format, I feel a little more detail is needed to interpret these results. While the methods described in the paragraph above are sufficient to get an idea of what is going on in the fieldwork, the stats need little more detail, particularly from line 119 where the use of survival analysis to analyse latency to arrive somewhat comes out of nowhere.

Line 101: In many ways, I feel it would be nicer to present these results in tables?

Line 110: I assume the dots here are a way of representing the raw data? It was slightly unclear to me at first read of the caption however. It might also be useful to remind the reader here how many trials each of these summary points consists of. I wonder also if violins or a vertical histogram might be a nice way of showing this distribution a little more transparently?

Line 201: This info might be better up in the intro, providing some extra backup to some of the

predictions. If leaving in place I also feel "specifically" is not quite right here, "in our case" or similar might fit better since you are talking about your particular study system.

Reviewer #3 (Remarks to the Author):

In this article the author's present a field based experimental study of the effects of group size and sex composition on social foraging in Trinidadian guppies. Using wild-caught guppies in natural habitats allowed them to examine social foraging in the presence of natural local environmental conditions which allows for a powerful examination of the effects of both sex and group size on both discovery and use of novel foraging patches as well as comparison to solitary foraging trials.

The article is very well written and presented with a clear and well-designed experimental setup. The results are detailed effectively and laid out in a logical fashion. The statistical methods are appropriately chosen, presented clearly and with all necessary detail and supplementary findings are all well documented. Relevant literature is cited throughout the manuscript. Both the introduction and discussion frame the study well in the broader context and highlight the key findings. Below are my minor comments on the article. Thank you for the lovely read.

Minor comments:

Results: As there are a considerable amount of model results presented in the same fashion (i.e. estimate +/- SE, N, x2, p) in each section I wonder if these might be presented more clearly in tables to ease the reading of the text without the flow interruption of the repeated model statistics. This is a preference-based comment, and others may have a different opinion on which is easiest to read.

Discussion: It might be interesting to touch on in the discussion some ideas of why males don't group more, given the findings here that they benefit equally or more from social foraging. For instance what potential trade-offs might outweigh these foraging gains and thus cause males to avoid social grouping more.

L123: reverse order of "did also" and remove "overall" so sentence reads "...also did not differ in how quickly they reached..."

L212-214: This sentence is quite long and a bit awkward to read, perhaps it can be re-written for clarity.

Something like: "We predicted that in both solitary and social conditions, females would show stronger foraging performance due to increased selective pressure from the strong link between resource availability and female fecundity."

L215: Here it states males and females did not differ in resource consumption when >6 individuals are present, however in the results (L167) it says did not differ at >4 individuals.

L216-217: I had a hard time following this sentence, is the interference competition mentioned here expected to occur in the males? I think the wording "constrained females to outperform" is confusing to me.

L221-223: Neophobia is listed in the introduction along with these other mechanisms discussed here, is there any previous data on this population if males and females differ in neophobic tendencies?

L281: Just out of curiosity, are there any possible explanations for the higher escape rate of females? It seems that as males are known to leave groups more often, male escape rate might have been expected to be higher.

REVIEWERS' COMMENTS:

Reviewer #1 (Remarks to the Author):

Review of Snijders et al. "Causal evidence for the adaptive benefits of social foraging in the wild"
I very much enjoyed reading this interesting manuscript and believe it would also be of board interest to biologists in general. Specifically, studies considering the costs and benefits of social foraging probably number in the thousands now, but nearly all of this work is either observations in the wild or captive-based experiments. So, to address this topic with a large-scale experiment like this in the wild will certainly now attract a lot of interest, and be an extremely useful addition to the literature. I found the writing to be excellent, the methods to a great combination of maintaining an intuitive approach to analysing the data as well as being intuitive and understandable for the reader, and the conclusions to follow well from the patterns presented in the data. The supplementary information is clearly formatted and referred to appropriately throughout the main text, and therefore will be useful for readers interested in specific aspects. As such, the suggestions below are aimed mainly at just bringing some attention to a few specific areas which could potentially be considered for some rather straight-forward changes:

RESPONSE: We highly appreciate these positive remarks and are grateful to the reviewer for the helpful suggestions.

L24: Change "...the more social and risk-averse sex,..." to "...the more social and risk-averse sex in this species,..."

RESPONSE: Changed accordingly

L27: I generally try to stay away from referring directly to specific statistical aspects within the abstract, and think it is often more intuitive to use descriptive terms, so I would suggest change "The slope of the latter relationship" to "The extent of the latter relationship"

RESPONSE: Changed accordingly

L40-42: This could be explained a bit further (e.g. an example of why this is the case, or when it has been the case in previous research)

RESPONSE: We have now added and changed the following sentences for clarification:
"For example, in observational studies individuals have the opportunity to self-select their social environment, which may lead to hidden correlations in group composition and group size. In addition, both group size and group performance may vary in response to variation in the environment (e.g. resource abundance), making it challenging to disentangle the driving factor. To allow for causal conclusions, experimental studies are conducted that directly manipulate the social environment [9–14]. Yet, these studies are predominantly done in captive settings and, therefore, likely to miss out on ecological relevance as they are unable to incorporate all of the local environmental pressures that shaped selection for sociality."

L98: Change "..., indicating..." to "...,potentially indicating..."

RESPONSE: We now changed "..., indicating..." to 'suggesting'

L102 Onwards: Probably no need to keep reiterating 'Est ± SE' (as I think the statement in line 101 will mean that readers will grasp that this is what the first two values separated by '±' are referring to)

RESPONSE: We have now moved most of the statistics to tables, thereby removing most reiterations.

L120-132: "Nnot censored/Ntotal = 1423/4841" is identical all the way through this paragraph and repeated exactly the same 8 times.

RESPONSE: We have now moved most of these statistics to Table 2, thereby removing the repeated mentioning of the sample size.

L135: Figure 2 is missing the 'A' and 'B' labels and the figure legend is missing info that 'A' is for females and 'B' is for males. Also, it isn't clear by this point in the MS why you'd graphically group females by different number groupings compared to the male number grouping size (i.e. Females 2-4 & 5-8 while Males is 3-4 & 7-8), so could benefit from a brief clarification in the legend.

RESPONSE: Figure is changed accordingly and the requested clarification is added to the start of the results and to all the figure legends.

L164-166: The pattern that the relationship between 'number of conspecifics' and 'bites taken' is stronger for males than females is interesting and clearly very strong in the data. But, I'm just curious whether this pattern is as strong if solitary individuals are excluded from this part of the analysis? Or is it strongly driven by the difference in solitary?

RESPONSE: This is an interesting question. We tested the model while excluding solitary individuals and still found a significant interaction between sex and number of conspecifics ($\chi^2 = 6.53$, $P = 0.01$, $N = 234$), so it appears that the relationship is not only driven by solitary individuals.

L205-206: Yes, very cool finding! Even within our single population of great tits (*Parus major*), males' personality is consistently related to various types of social behaviour, including network position (Aplin et al., 2013) and social pair-bonding strength (Firth et al., 2018) during winter foraging, and social assortment in territories during breeding (Johnson et al., 2017), while females' personality is not related to any of these components of sociality. Not sure any of this information will be useful to the authors (indeed, probably more relevant to keep their own examples focussed on fish) but thought I should share as its really interesting to see such obvious effects with sex differences here in their experimental system!

RESPONSE: We very much appreciate that the reviewer shares the same enthusiasm for these findings. The topic of when we may or may not expect sex-differences in social behaviour and social benefits surely deserves to be more thoroughly explored across experimental field systems and we very much welcome further discussion.

L232-233: This the statement meant generally across species? Or specific to this system/species?

RESPONSE: We now realize that this statement was too general and changed it to 'many male animals'

L241-242: It is a bit hard to imagine what exactly is meant by "experimentally lower the energy reserves of individual males" in terms of the proposed methodology. Have any previous studies been successful in this kind of manipulation?

RESPONSE: We apologize for not being clearer. We have now added a concrete example "e.g. by temporarily restricting access to food"

L291: Change “protocol similar to [22,42] ” To “protocol similar to prior work [22,42]”

RESPONSE: Changed accordingly

L316-318: These sample sizes are quite important and should probably be mentioned in the earlier ‘Results’ section in the appropriate sections if there is room to do so.

RESPONSE: We agree and have moved this information to the results section.

I hope these comments are useful for the authors, and I congratulate them on an excellent piece of research.

Best wishes,

Josh Firth (please note, I sign all my reviews)

Reviewer #2 (Remarks to the Author):

I found this to be a very interesting and nicely written paper with an impressive dataset to back it up. Manipulating the size of groups in the wild and then recording detailed foraging data is a good way to get at some of the key questions about group foraging.

I think the Results section needs a little bit more methodological detail given the format. I also think tables would be more readable than presenting the results in sentences. Other than this, just a few minor comments/suggestions.

RESPONSE: We are grateful for the positive remarks and helpful suggestions made by the reviewer.

Line 46: This is possibly needlessly picky, but would you include removal experiments here? I know that these probably do not subsequently focus on foraging behaviour, but I feel they somewhat fall into the category of manipulating number of conspecifics in the wild.

RESPONSE: The reviewer makes a valid point. Such studies have been invaluable and should be acknowledged. We now mention this type of experiment as a notable exception.

Line 95: In this sort of results first format, I feel a little more detail is needed to interpret these results. While the methods described in the paragraph above are sufficient to get an idea of what is going on in the fieldwork, the stats need little more detail, particularly from line 119 where the use of survival analysis to analyse latency to arrive somewhat comes out of nowhere.

RESPONSE: To give more methodological detail earlier on, we now added a short section at the start of the results section mentioning the number of assessed trials (following Reviewer 1’s suggestion) and the basic statistical approach.

Line 101: In many ways, I feel it would be nicer to present these results in tables?

RESPONSE: We agree and have moved the summary statistics for the four main models to tables.

Line 110: I assume the dots here are a way of representing the raw data? It was slightly unclear to me at first read of the caption however. It might also be useful to remind the reader here how many

trials each of these summary points consists of. I wonder also if violins or a vertical histogram might be a nice way of showing this distribution a little more transparently?

RESPONSE: The dots are actually a way to visualise variation in the predicted values. We choose to visualise the predicted values (also in Figure 3) because the raw data on which we conducted the analysis are just ones and zeroes and therefore unfortunately not very informative. We now made this clearer and added more information about the sample sizes to all the figure legends.

Line 201: This info might be better up in the intro, providing some extra backup to some of the predictions. If leaving in place I also feel “specifically” is not quite right here, “in our case” or similar might fit better since you are talking about your particular study system.

RESPONSE: We agree that this information is also relevant for the introduction, but also the discussion. We decided to keep the information in the introduction a bit more general, i.e. “previous work showed that male guppies are generally less social [22,44–48]” and explore this information underlying our predictions in more detail in the discussion. Following the reviewer’s suggestion, we have now changed “specifically” to “In this species”.

Reviewer #3 (Remarks to the Author):

In this article the author’s present a field based experimental study of the effects of group size and sex composition on social foraging in Trinidadian guppies. Using wild-caught guppies in natural habitats allowed them to examine social foraging in the presence of natural local environmental conditions which allows for a powerful examination of the effects of both sex and group size on both discovery and use of novel foraging patches as well as comparison to solitary foraging trials.

The article is very well written and presented with a clear and well-designed experimental setup. The results are detailed effectively and laid out in a logical fashion. The statistical methods are appropriately chosen, presented clearly and with all necessary detail and supplementary findings are all well documented. Relevant literature is cited throughout the manuscript. Both the introduction and discussion frame the study well in the broader context and highlight the key findings. Below are my minor comments on the article. Thank you for the lovely read.

RESPONSE: We thank the reviewer for the positive words and helpful suggestions.

Minor comments:

Results: As there are a considerable amount of model results presented in the same fashion (i.e. estimate +/- SE, N, χ^2 , p) in each section I wonder if these might be presented more clearly in tables to ease the reading of the text without the flow interruption of the repeated model statistics. This is a preference-based comment, and others may have a different opinion on which is easiest to read.

RESPONSE: We agree with the comment (as also suggested by reviewer 2) and have moved most of this information to tables.

Discussion: It might be interesting to touch on in the discussion some ideas of why males don’t group more, given the findings here that they benefit equally or more from social foraging. For instance what potential trade-offs might outweigh these foraging gains and thus cause males to avoid social grouping more.

RESPONSE: We thank the reviewer for this suggestion. This is an interesting point and we now added the following to the final paragraph in the discussion: "That male guppies are generally, but not always^{22,42}, found to be less social, suggests that foraging benefits may not outweigh the costs of being social for males or, alternatively, that we as researchers may need to be more precise in our definition of 'social' (e.g. preferred associations *versus* time spent social *versus* number of unique social contacts). In addition, the documented sex differences in guppy sociality may be mainly driven by females following sex-specific benefits outside the foraging context (e.g. avoiding predation). Investigations into the individual states and ecological characteristics that modulate the individual costs and benefits of sociality in the wild, both within a foraging context and beyond, offer fruitful avenues for future research."

L123: reverse order of "did also" and remove "overall" so sentence reads "...also did not differ in how quickly they reached..."

RESPONSE: Changed accordingly

L212-214: This sentence is quite long and a bit awkward to read, perhaps it can be re-written for clarity.

Something like: "We predicted that in both solitary and social conditions, females would show stronger foraging performance due to increased selective pressure from the strong link between resource availability and female fecundity."

RESPONSE: Changed accordingly

L215: Here it states males and females did not differ in resource consumption when >6 individuals are present, however in the results (L167) it says did not differ at >4 individuals.

RESPONSE: We apologize for the confusion. Because for males there were only batches of 7 and 8 fish in the category 5-8 fish while for females there were batches of 5, 6, 7 and 8 fish, we used >4 when speaking about both sexes and >6 when speaking about males. We have now clarified this by rephrasing the sentence in the results section to: "but males and females in a pool with at least four (females) or six (males) additional conspecifics did not differ". As the sentence in the discussion refers specifically to males, we kept the sentence as it is, i.e. "males reached comparable levels of total resource consumption when at least six other same-sex conspecifics were present." In addition, we now stress in each of the main figure legends that there were no batches of 5 and 6 fish for males.

L216-217: I had a hard time following this sentence, is the interference competition mentioned here expected to occur in the males? I think the wording "constrained females to outperform" is confusing to me.

RESPONSE: We have now rephrased this sentence more clearly: "Higher interference competition at the patch [28] may have partly constrained fish to take more bites and consequently prevented females to outperform males in highly social conditions"

L221-223: Neophobia is listed in the introduction along with these other mechanisms discussed here, is there any previous data on this population if males and females differ in neophobic tendencies?

RESPONSE: Harris et al (2010), for example, found that wild-caught male guppies from low predation populations in Trinidad (similar to but not identical to our study population) emerged quicker into a novel environment than females. Unfortunately, some other excellent studies on neophobia in Trinidadian guppies (e.g. Brown et al., 2013) only focussed on females. Generally, little is known

about sex differences in neophobia (Crane et al., 2017) and we are not aware of a study specific to our study population, but this would surely be interesting for future research. We now added neophobia to the list of potential mechanisms in the discussion: "Increase in perceived safety [11] or decrease in neophobia [13] are also unlikely mechanisms, as we then would expect females (the more risk-averse and possibly more neophobic sex in guppies [44,49–51]) to show the strongest improvement in social versus asocial conditions"

L281: Just out of curiosity, are there any possible explanations for the higher escape rate of females? It seems that as males are known to leave groups more often, male escape rate might have been expected to be higher.

RESPONSE: Indeed we also expected males to be the ones (if any) to escape more often and we were quite surprised by this finding. Some highly speculative explanations include females possibly being able to jump higher/further or being more cognitively flexible in relation to barrier avoidance (e.g. Lucon-Xiccato & Bisazza, 2017) and consequently being more successful escape artists than males.